# Uncertainty-informed deep learning models enable high-confidence predictions for digital histopathology

James M. Dolezal [1], Andrew Srisuwananukorn [2], Dmitry Karpeyev[3], Siddhi Ramesh[1], Sara Kochanny[1], Brittany Cody[4], Aaron S. Mansfield [5], Sagar Rakshit[5], Radhika Bansal [5], Melanie C. Bois[6], Aaron O. Bungum[7], Jefree J. Schulte [8], Everett E. Vokes[1], Marina Chiara Garassino[1], Aliya N. Husain[4] & Alexander T. Pearson [1]✉

A model's ability to express its own predictive uncertainty is an essential attribute for maintaining clinical user confidence as computational biomarkers are deployed into real-world medical settings. In the domain of cancer digital histopathology, we describe a clinically-oriented approach to uncertainty quantification for whole-slide images, estimating uncertainty using dropout and calculating thresholds on training data to establish cutoffs for low- and high-confidence predictions. We train models to identify lung adenocarcinoma vs. squamous cell carcinoma and show that high-confidence predictions outperform predictions without uncertainty, in both cross-validation and testing on two large external datasets spanning multiple institutions. Our testing strategy closely approximates real-world application, with predictions generated on unsupervised, unannotated slides using predetermined thresholds. Furthermore, we show that uncertainty thresholding remains reliable in the setting of domain shift, with accurate high-confidence predictions of adenocarcinoma vs. squamous cell carcinoma for out-of-distribution, non-lung cancer cohorts.

Deep learning models have shown incredible promise in the field of digital histopathology. Within the domain of oncology, deep neural networks enable rapid and automated morphologic segmentation[1–6], tumor classification[7–10], and grading[11–14], as well as prognostication[15–17] and treatment response prediction[18–21]. While artificial intelligence (AI) methods may hold the key to developing advanced tools capable of out-performing human experts for these tasks, the unpredictable nature of deep learning models on data outside the training distribution impedes clinical application. The observation that deep learning model performance deteriorates when applied to data

falling outside the training distribution, a phenomenon known as domain shift[22,23], raises the burden of proof for clinical application. Assessing performance on external test sets is a crucial component of evaluating potential utility of any deep learning model, but practical limitations in the availability and diversity of clinical data challenges our ability to accurately predict how well a model will generalize to other institutions and patient populations. This unpredictable nature limits our capacity to reliably ascribe confidence to a model's performance and constitutes a principal component behind the reluctance to deploy deep learning models as clinical decision

[1]Section of Hematology/Oncology, Department of Medicine, University of Chicago Medical Center, Chicago, IL, USA. [2]Tisch Cancer Institute, Icahn School of Medicine at Mount Sinai, New York, NY, USA. [3]DV Group, LLC, Chicago, IL, USA. [4]Department of Pathology, University of Chicago, Chicago, IL, USA. [5]Division of Medical Oncology, Mayo Clinic, Rochester, MN, USA. [6]Department of Laboratory Medicine and Pathology, Mayo Clinic, Rochester, MN, USA. [7]Divisions of Pulmonary Medicine and Critical Care, Mayo Clinic, Rochester, MN, USA. [8]Department of Pathology and Laboratory Medicine, University of Wisconsin at Madison, Madison, WN, USA. ✉e-mail: apearson5@medicine.bsd.uchicago.edu

support tools, particularly if a model aims to affect treatment decisions for patients.

Over the past several years, there has been a growing awareness of the need for better estimates of confidence and uncertainty within medical applications of AI[24–26]. Many domains of routine clinical practice incorporate measures of uncertainty; within the field of pathology, for example, it is not uncommon for a diagnostic study to lack sufficient material or possess morphologic ambiguity that precludes reliable diagnosis. Most deep learning applications within digital pathology, however, do not include the ability to assess case-wise uncertainty, rendering predictions regardless of histologic ambiguity. For some clinical applications, it may be permissible for a deep learning model to abstain from generating predictions if it can be known that the prediction is low-confidence. Such a model may still prove useful if results that fall into a higher confidence range are actionable.

Several techniques have been developed to estimate uncertainty from deep learning models. Most uncertainty quantification (UQ) methods reformulate model output from a single prediction to a distribution of predictions for each sample. The most common UQ method involves randomly dropping out a proportion of nodes within the model architecture when generating predictions, a technique known as Monte Carlo (MC) dropout. Dropout layers are included in the model architecture and enabled during both training and inference. During inference, a single input sample undergoes multiple forward passes in the network to generate a distribution of predictions. The standard deviation of such a distribution has been shown to an approximate sampling of the Bayesian posterior and has been used to estimate uncertainty[27,28]. A second method for UQ estimation involves the use of deep ensembles. Deep ensembles generate a distribution of predictions for input samples by training several separate deep learning models of the same architecture, resulting in multiple predictions for each input[29]. Hyper-deep ensembles are an extension of this method where each model is trained with different set of hyperparameters[30]. In both cases, variance or entropy of model output can be used for uncertainty estimation. A third UQ method is test time augmentation (TTA), where a given input sample undergoes a set of random transformations, with predictions generated for each perturbed image. Prediction variance or entropy can be used for uncertainty estimation[31].

The utility of these UQ methods has been explored for various applications in digital histopathology, including segmentation[1–3,6,32], classification[32–38], and dataset curation[32,38]. In general, high uncertainty is associated with misclassification or poorer quality segmentation, a phenomenon potentially exploitable for isolating a subset of high-confidence predictions. However, consistent with previous observations in the broader machine learning literature[22,23,39,40], uncertainty estimates were susceptible to domain shift when applied to external datasets[36,37], raising concerns about generalizability.

A limitation in the above studies is the method of assessing the reliability of uncertainty estimates in the face of observed domain shift. Several groups explored UQ thresholds to enable high-confidence predictions on low-uncertainty data, abstaining from predictions for high-uncertainty data[33–35,37]. In each of these approaches, however, thresholds were manually determined when the distribution of uncertainty in validation data was known, constituting a form of data leakage. With uncertainty distributions susceptible to domain shift, it is critical that thresholds are determined on only training data. Additionally, in cases where uncertainty was estimated for a classification task, uncertainty estimates were provided only for smaller subsections of a slide and not whole-slide images (WSI). Uncertainty estimates for WSIs will provide significant advantages when applied for patient-level predictions in a clinical context.

Reliable, patient-oriented estimation of uncertainty is paramount for building actionable deep learning models for clinical practice.

In this work, we describe a clinically-oriented method for determining slide-level confidence using the Bayesian approach to estimating uncertainty, with uncertainty thresholds determined from training data. We test our uncertainty thresholding method on deep convolutional neural network (DCNN) models trained to predict the histologically well-defined outcome of lung adenocarcinoma vs. squamous cell carcinoma and use two large, external datasets for robust validation. The key contributions of this work include a method for estimating slide-level uncertainty for WSIs which provides potentially actionable information at the patient level, a nested cross-validation uncertainty thresholding strategy immune to validation data leakage, assessment of the amount of training data necessary for actionable uncertainty estimates, and robust external evaluation of our uncertainty thresholding strategy on two large datasets comprised of data from multiple institutions.

## Results

### Uncertainty thresholding improves accuracy for high-confidence predictions

A total of 276 standard (non-UQ) and 504 UQ-enabled DCNN models based on the Xception architecture were trained to discriminate between lung squamous cell carcinoma and lung adenocarcinoma using varying amounts of data from TCGA. UQ models were trained according to the experimental strategy illustrated in Fig. 1. Resulting cross-validated AUROCs from standard, non-UQ models trained for the binary categorization task are shown in Fig. 2a. At maximum dataset size (941 slides), cross-validation AUROC among non-UQ models is $0.960 \pm 0.008$. UQ-enabled models were trained in cross-validation for dataset sizes ≥ 100 slides, with uncertainty and prediction thresholds determined through nested cross-validation. AUROCs within the high-confidence UQ cohorts are greater than the non-UQ AUROCs for all dataset sizes ≥ 100 slides with $\alpha = 0.05$, reaching $0.981 \pm 0.004$ at maximum dataset size ($P < 0.001$) (Fig. 2b). The proportion of slides classified as high-confidence in each validation dataset ranged from 79–94% (Fig. 2c). Cross-validation was also performed at the maximum dataset size for four other architectures (InceptionV3, ResNet50V2, InceptionResNetV2, EfficientNetV2M) to assess generalizability for other network designs, and AUROCs from high-confidence predictions were higher than AUROCs from non-UQ models in all cases (Supplementary Fig. 1).

Consistent with expectations, cross-validated AUROC decreases as classes are increasingly imbalanced, with the deterioration in AUROC partially alleviated by increasing dataset size (Fig. 3a). High-confidence predictions in the UQ cohorts outperform models without UQ with a class imbalance ratio of 1:3 and dataset size ≥ 200 slides, but high-confidence predictions do not outperform standard non-UQ models for highly imbalanced datasets with a 1:10 outcome ratio (Fig. 3b, c).

When evaluated on the CPTAC dataset, high-confidence predictions from UQ models outperform non-UQ models with respect to AUROC, accuracy, and Youden's index, with the effect most prominent for training dataset sizes ≥ 200 slides (Fig. 4a, c). Without UQ, a model trained on the full TCGA training set has a patient-level AUROC of 0.93, accuracy of 85.3%, sensitivity of 91.1%, and specificity of 79.8%. With UQ, the high-confidence cohort from the model trained at maximum dataset size reaches a patient-level AUROC of 0.99, accuracy of 97.5%, and sensitivity/specificity of 98.4 and 96.7%, respectively. Across all training dataset sizes, 66–100% of patients in the CPTAC cohort are classified with high-confidence. For comparison, the best-performing model among the low-confidence cohort has an AUROC of 0.75. Plots associating slide-level uncertainty with probability of misclassification are shown in Supplementary Fig. 2.

On an institutional dataset from Mayo Clinic, the same pattern of improved predictions with UQ is observed, but with higher baseline performance in the non-UQ model (Fig. 4b, c). Without UQ, the model

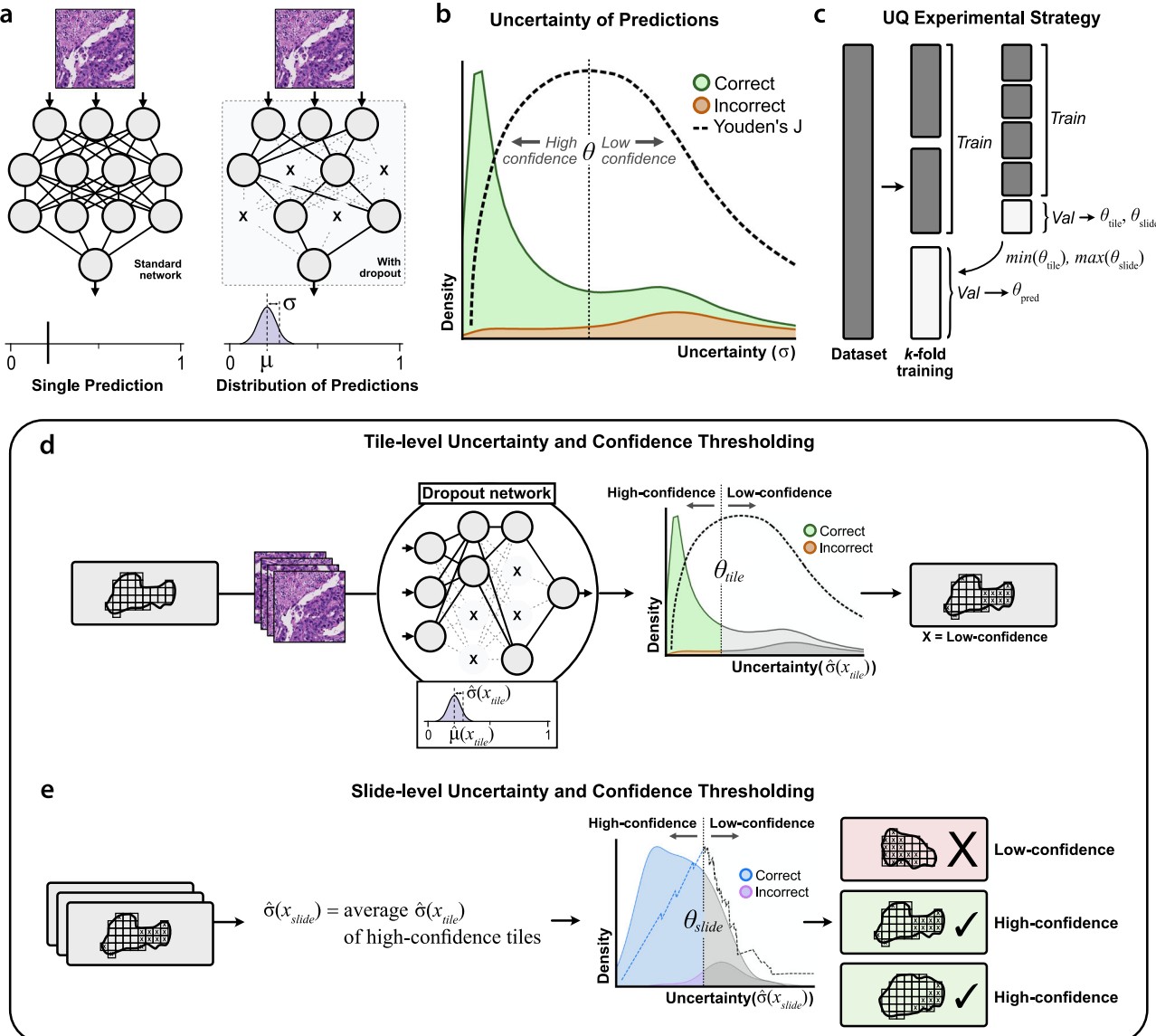

**Fig. 1 | Estimation of uncertainty and confidence thresholding. a** With standard deep learning neural networks, a single image yields a single output prediction. When dropout is enabled during inference, predictions for a single image will vary based on which nodes are randomly dropped out. To estimate tile-level uncertainty, images undergo 30 forward passes through the network, resulting in a distribution of predictions. The mean of this distribution, $\hat{\mu}(x_{tile})$, represents the final tile-level prediction, and the standard deviation, $\hat{\sigma}(x_{tile})$, represents the tile-level uncertainty. **b** When uncertainty quantification (UQ) methods are used, incorrect predictions are associated with higher uncertainties than correct predictions[32–38]. From a given distribution of tile- or slide-level uncertainties, we determine the uncertainty threshold $\theta$ which optimally separates correct and incorrect predictions by maximizing Youden's index ($J$). Predictions with uncertainty below this threshold are high-confidence, and all others are low-confidence. **c** To prevent data leakage and overfitting, optimal tile- and slide-level uncertainty thresholds are determined through nested cross-validation within training folds. **d** Schematic for calculating tile-level uncertainty and confidence. The optimal tile-level uncertainty threshold $\theta_{tile}$ is calculated from a given validation dataset. Tiles from the dataset are separated into high- and low-confidence by whether the tile-level uncertainty falls below or above $\theta_{tile}$, respectively. **e** Schematic for slide-level uncertainty and confidence. Slide-level uncertainty is defined as the average uncertainty among high-confidence tiles for a given slide. The optimal slide-level uncertainty threshold $\theta_{slide}$ is found and used to classify slides as high- and low-confidence.

trained on the full TCGA dataset has an AUROC of 0.98, an accuracy of 94.1%, sensitivity of 82.5%, and specificity of 97.3%. With UQ, the high-confidence cohort from this same dataset size reaches an AUROC of 1.0, accuracy of 100%, and sensitivity/specificity of 100% and 100%, respectively. Across all training dataset sizes, 70.9–94.6% of patients receive a high-confidence prediction. The low-confidence cohort has an AUROC of 0.94 at the maximum dataset size.

## Uncertainty thresholding generalizes to out-of-distribution data
Using a UQ model trained on the full TCGA lung adenocarcinoma vs. squamous cell carcinoma dataset ($n = 941$), predictions were generated for all non-lung squamous and adenocarcinoma cancers in TCGA to test prediction and uncertainty thresholding performance on out-of-distribution slides from a different tissue origin (Fig. 4d). Predictions were generated for 700 squamous cell cancers and 2456 adenocarcinomas. The squamous cell cohort included primary sites of cervix (CESC), esophagus (ESCA), and head and neck (HNSC). The adenocarcinoma cohort included primary sites of breast (BRCA), cervix (CESC), colon (COAD), esophagus (ESCA), kidney (KIRP), ovary (OV), pancreas (PAAD), prostate (PRAD), stomach (STAD), thyroid (THCA), and uterus (UCEC). Using a non-UQ model trained on lung cancer, 98.6% of non-lung squamous cell cancers were

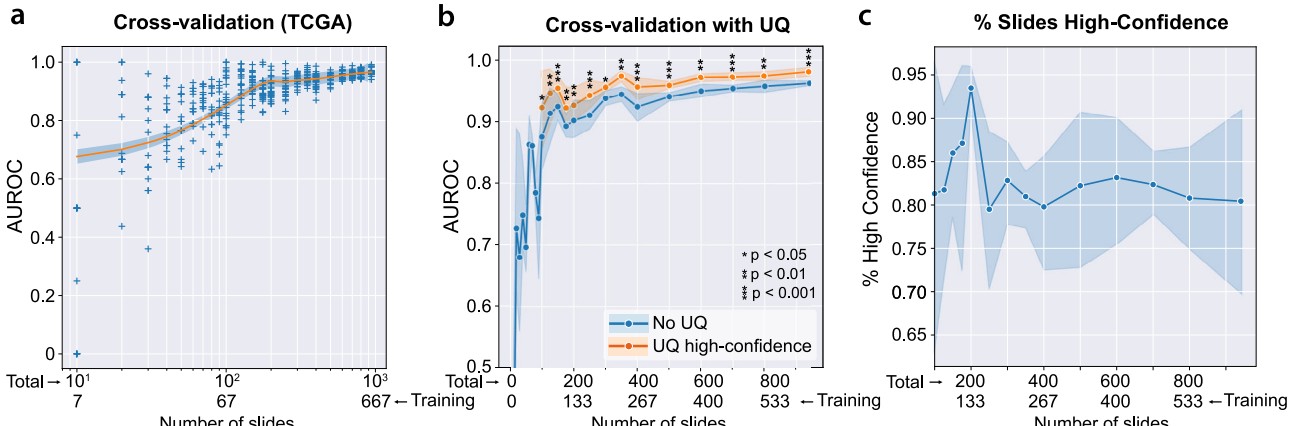

**Fig. 2 | Uncertainty thresholding yields improved accuracy for high-confidence predictions. a** Cross-validation area under receiver operator curve (AUROC) from 276 models trained on The Cancer Genome Atlas (TCGA) to predict lung adenocarcinoma vs. squamous cell carcinoma, using varying amounts of training data. AUROC at the largest dataset size is 0.960 ± 0.008. The regression line shown is a Loess estimate, shown with a 95% confidence interval obtained through bootstrapping with center line representing the mean. **b** For datasets larger than 100 slides, uncertainty quantification (UQ) was performed to identify low and high-confidence predictions, with the high-confidence predictions shown in comparison to all predictions. AUROCs from high-confidence UQ cohorts are significantly

higher than those without UQ for all dataset sizes ≥ 100 slides with α = 0.05. Statistical comparisons were made using one-sided, paired *t*-tests. The shaded interval represents the AUROC 95% confidence interval at each dataset size, with the center line representing the mean. **c** Across all cross-validation experiments with UQ, 81.9% of predictions are classified as high-confidence. At each dataset size, the average percent of validation predictions classified as high-confidence is 82.8%, with a median of 84.6% (43.8–100%). The shaded interval represents the 95% confidence interval at each dataset size, with the center line representing the mean. Source data are provided as a Source Data file.

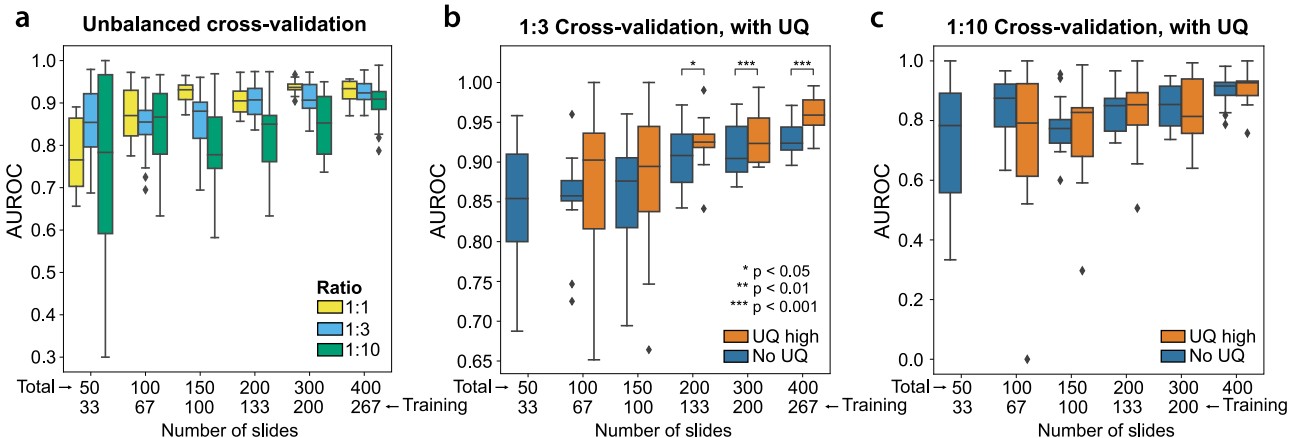

**Fig. 3 | Uncertainty thresholding for unbalanced data. a** Datasets with unbalanced ratios of outcomes yield inferior cross-validation performance compared to balanced datasets at equivalent dataset sizes. Boxplots summarize results from 12 models trained at each dataset size for the 1:1 ratio experiments and 24 models trained at each dataset size for the 1:3 and 1:10 ratio experiments. **b** Datasets unbalanced at 1:3 ratios also experience improvement from the use of uncertainty quantification (UQ), with area under receiver operator curves (AUROCs) improved in the high-confidence cohorts. For dataset sizes of 200, 300, and 400 slides, *p* = 0.018, 0.00037, and <0.0001, respectively. Statistical comparisons were made

using one-sided, paired *t*-tests. Boxplots summarize results from 12 models trained at each dataset size. **c** Datasets unbalanced at 1:10 ratios do not benefit from UQ, with similar performance in high-confidence cohorts compared to all predictions. Boxplots summarize results from 12 models trained at each dataset size. For all boxplots, center line represents the median (50th percentile), lower and upper box bounds represent interquartile range (25th–75th percentile), and minimum (lower whisker) and maximum (upper whisker) bounds extend to furthest datapoint up to 1.5 times the interquartile range, with outliers shown as diamonds. Source data are provided as a Source Data file.

correctly predicted to be squamous cell, and 76.3% adenocarcinomas were correctly predicted to be adenocarcinoma. With uncertainty thresholding, 66.0% (462 of 700) of squamous cell cancers yielded high-confidence predictions which were 99.8% accurate. In the adenocarcinoma cohorts, 59.4% (1458 of 2456) slides yielded high-confidence predictions with an accuracy of 95.2%.

Predictions were also generated for 4015 non-lung, non-adenocarcinoma, non-squamous tumors, comprising a set of slides for which there should be no correct diagnosis (Supplementary Fig. 3). Of these slides, 3153 (78.5%) were reported as low-confidence. Of the remaining slides with high-confidence predictions, 412 (10.3%) were predicted to be squamous cell, and 450 (11.2%) were predicted to be adenocarcinomas.

## Areas of high uncertainty correlate with histologic ambiguity

A sample adenocarcinoma WSI from the CPTAC external evaluation dataset is shown in Fig. 5 with heatmaps of all predictions, tile-level uncertainty, and high-confidence predictions. Attention is given to the lowest-uncertainty and highest-uncertainty image tiles, demonstrating that nearly all high-confidence image tiles show clear, unambiguous adenocarcinoma morphology as determined by two expert pathologists. Although the low-confidence image tiles lack the clear glandular structures seen among the high-confidence images, some low-confidence images possess features associated with adenocarcinoma, including micropapillary and lepidic morphologies. Quantitative pathologist assessment of these features is shown in Supplementary Fig. 4.

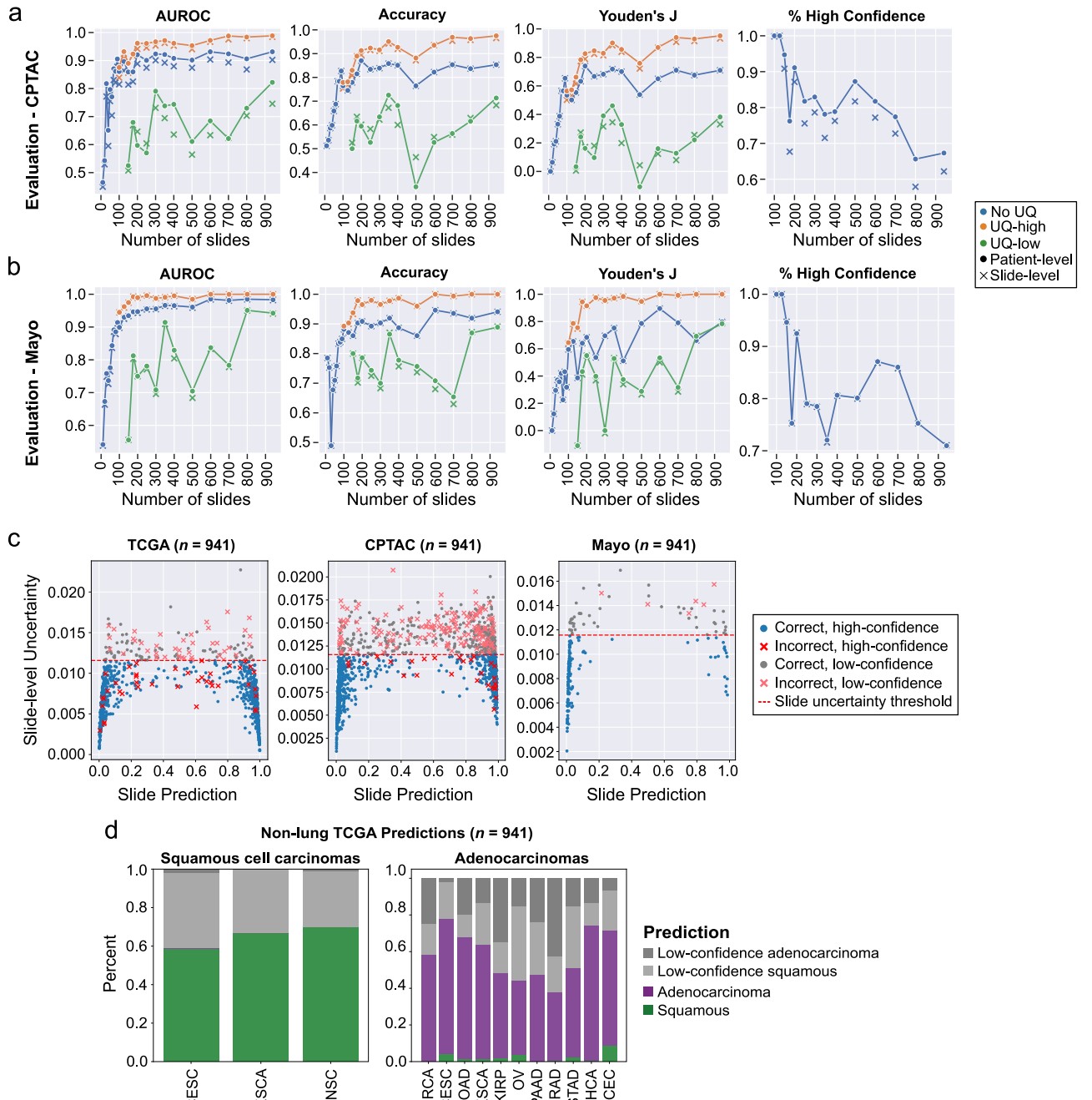

**Fig. 4 | Uncertainty thresholding improves predictions on external datasets and in the setting of domain shift. a** Models trained on The Cancer Genome Atlas (TCGA) at varying dataset sizes were validated on lung adenocarcinomas and squamous cell carcinomas from the Clinical Proteomic Tumor Analysis Consortium (CPTAC). Patient-level metrics are shown with the dotted lines, and slide-level metrics are shown with Xs. Area under receiver operator curve (AUROC), accuracy, and Youden's J are all improved in the high-confidence uncertainty quantification (UQ) cohorts. The proportion of patients and slides reported as high-confidence is shown in the last panel. **b** Evaluation results on an institutional dataset of 150 adenocarcinomas and 40 squamous cell carcinomas. Overall performance is higher than on CPTAC, but the same pattern of superior performance in the high-confidence UQ cohorts remains. Fewer slides were excluded as low-confidence in this dataset. **c** The relationship between slide-level uncertainty and slide prediction is shown for the aggregated TCGA cross-validation results, CPTAC predictions, and Mayo predictions for the experiment trained on the full TCGA dataset (number of slides = 941). Predictions near 0 are consistent with adenocarcinoma, and predictions near 1 are consistent with squamous cell carcinoma. The red dotted line indicates the slide-level uncertainty threshold. **d** For this same model, predictions were then generated for 700 domain-shifted, non-lung squamous cell cancers and 2456 non-lung adenocarcinomas, with both high-confidence and low-confidence predictions shown. Predictions from bladder (BLCA) and liver (LIHC) cohorts are not shown due to low sample sizes (*n* < 2). With uncertainty thresholding, classification accuracy in high confidence cohorts for non-lung squamous cell cancers and non-lung adenocarcinomas is 99.8 and 95.2%, respectively. Source data are provided as a Source Data file.

Activation and mosaic maps generated from predictions on the CPTAC cohort are shown in Fig. 6. Four regions are manually highlighted to visualize subsections with distinct morphologies. Image tiles with high-confidence adenocarcinoma predictions are localized near area 1. Nearly all 36 image tiles in Area 1 have clear adenocarcinoma morphology, with gland formation and some with mucin. Only one tile in this section shows ambiguity (row 6, col 6). Area 2 marks an area enriched with high-confidence predictions of squamous cell

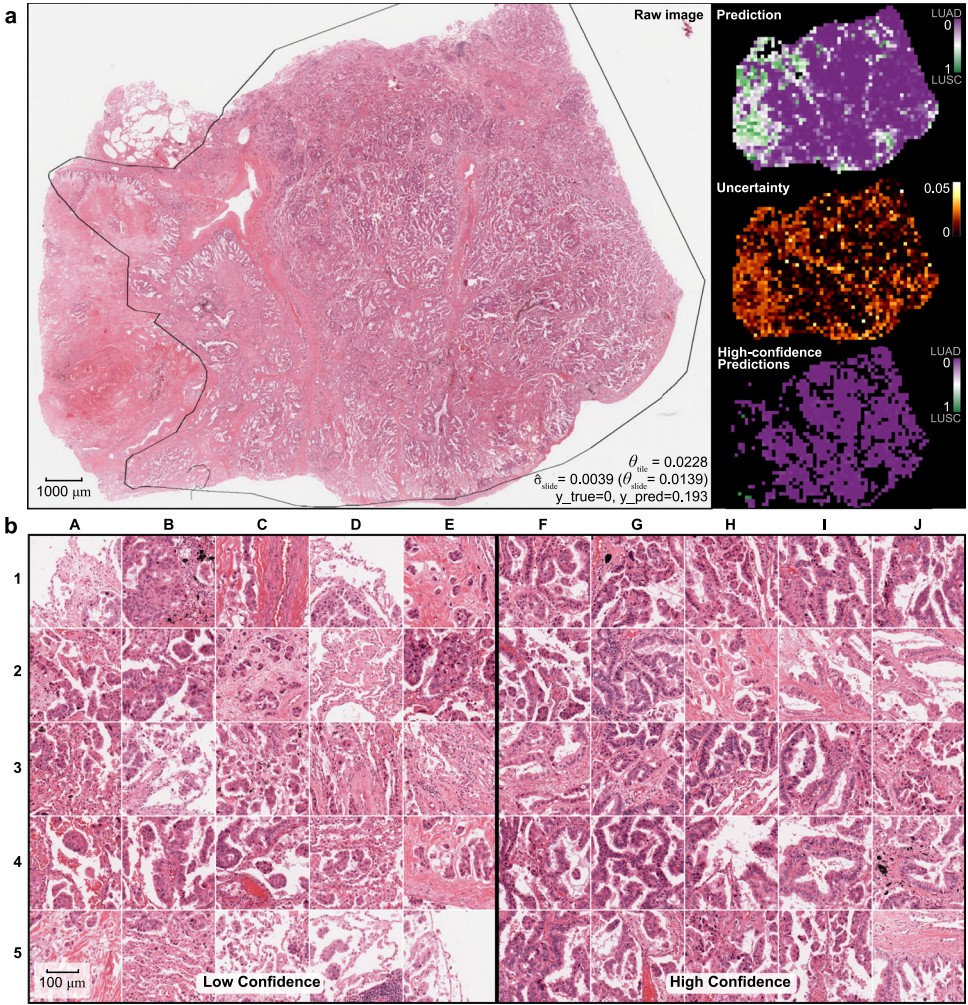

**Fig. 5 | Visualization of uncertainty and confidence in a validation slide. a** One representative adenocarcinoma from the CPTAC evaluation dataset is shown, outlined with a pathologist-annotated ROI for reference only. Predictions were generated for all tiles from this whole-slide image excluding only background whitespace. Tile-level predictions from a model trained on the full TCGA dataset is shown in the top-right, with purple indicating prediction near 0 (consistent with the correct diagnosis of adenocarcinoma), and green indicating predictions near 1 (squamous cell carcinoma). Tile-level uncertainty is shown in the middle-right panel. The bottom right shows only high-confidence tile-level predictions using the predetermined uncertainty threshold, demonstrating that virtually all high-confidence predictions are consistent with the correct diagnosis. **b** Twenty-five of the lowest confidence tiles are shown on the left, and the 25 highest confidence tiles are shown on the right. All high-confidence tiles show clear adenocarcinoma morphology. Glandular structures are dominant among these tiles, although tiles I2 and J2 appear lepidic in nature, and H3 shows micropapillary morphology. In contrast, the majority of low-confidence tiles lack clear glandular structures. Lepidic morphology is seen in B4, C4, D5, and E5, and micropapillary structures can be found in E1, C2, B3, C4, D4, and E4. All image tiles have a width of 302 μm.

carcinoma. Nearly all 36 images in Area 2 appear clearly squamous, some with basaloid morphology and/or keratinization. Only one image is not clearly squamous, containing mostly inflammation (row 6 col 5).

Area 3 highlights a section of low-confidence images near the high-confidence decision boundary. The majority of these low-confidence images contain sections of tumor with ambiguous morphology. Three tiles have micropapillary morphology (row 3 col 1, row 4 col 2, row 6 col 2), one has possible keratinization suggesting squamous morphology (row 1 col 4) and one is clearly squamous (row 4 col 5). Area 4 is a section of the mosaic map containing low-confidence images opposite the high-confidence adenocarcinomas and squamous cell carcinomas and distant from the decision boundary. All image tiles in area 4 appear benign, containing mostly background lung and stroma with only minute sections of tumor.

## UQ thresholding identifies decision-boundary uncertainty

A separate class-conditional generative adversarial network (GAN) model was trained using StyleGAN2 to generate LUSC, LUAD, and intermediate synthetic images approximately near the decision boundary (Fig. 7a and Supplementary Fig. 5). Predictions of 1000 synthetic GAN-LUAD, GAN-LUSC, and GAN-Intermediate images were calculated from a classification model trained on the full lung cancer TCGA dataset (Fig. 7b). These predictions were generally consistent with the synthetic image labels, particularly when the predictions were high-confidence. GAN-Intermediate images show an even spread of predictions along the adenocarcinoma/squamous cell carcinoma output spectrum, validating the morphologically intermediate nature of the images.

Cross-validation models were trained on TCGA with up to 500 real lung cancer slides and increasing numbers of GAN-Intermediate synthetic slides using random (LUAD v. LUSC) labels. Models trained on 500 real slides and no GAN-Intermediate slides have an average AUROC of 0.939 ± 0.005. Cross-validation AUROC progressively degrades with the addition of neutral, randomly labeled synthetic slides, with average AUROC at the maximum dataset size decreasing to 0.811 ± 0.024 when 50% of training and validation data included GAN-Intermediate slides (Fig. 7c).

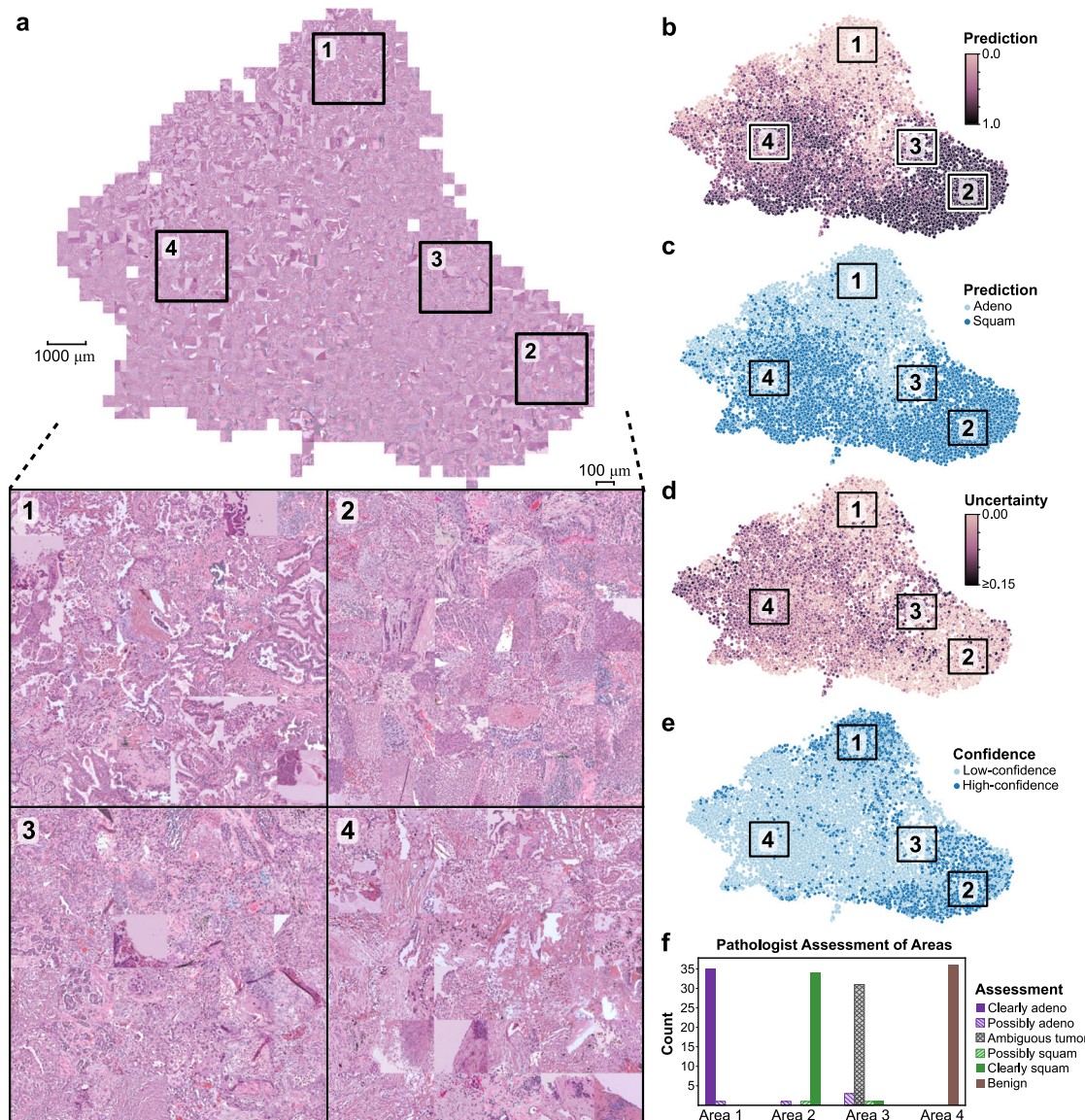

**Fig. 6 | Activation maps of external predictions highlighting areas of uncertainty. a** Penultimate hidden layer activations were calculated for image tiles in the CPTAC dataset and plotted with UMAP. Corresponding images for tiles were then overlaid in a grid-wise fashion to create the shown mosaic map. Four areas of interest are highlighted for magnified display, each with a total of 36 image tiles: (1) high-confidence adenocarcinoma predictions, (2) high-confidence squamous cell predictions, (3) low-confidence images at the boundary between the high-confidence adenocarcinoma and squamous cell predictions, and (4) low-confidence images far away from the high-confidence class boundary. **b** Tile-level predictions are shown overlaid on the UMAP, scaled from 0 (adenocarcinoma) to 1 (squamous cell). **c** Discretized tile-level class predictions. **d** Tile-level uncertainty. **e** Tile-level confidence, where high-confidence image tiles are defined as having uncertainty below $\theta_{tile}$. **f** Two pathologists reviewed the 144 images shown in Areas 1–4, and pathologic assessment of these images is summarized in the shown bar chart. All image tiles have a width of 302 μm. Source data are provided as a Source Data file.

UQ models were then trained at the maximum dataset size to test the ability of UQ to account for the non-informative synthetic images. Average AUROC for high-confidence predictions ranged between 0.945 and 0.966 despite the presence of increasing amounts of GAN-Intermediate slides (Fig. 7d). For these models, the proportion of GAN-Intermediate slides that yielded high-confidence predictions ranged from 0–0.8%, whereas the proportion of real LUAD and LUSC slides with high-confidence predictions ranged between 70.6 and 93.0% (Fig. 7e).

## Discussion

As with many cancers, accurate diagnosis is the critical first step in the management of non-small cell lung cancer, with management pivoting upon classification into squamous cell carcinoma or adenocarcinoma. As deep learning models are explored for these crucial steps in clinical

diagnostics, it is imperative that estimations of uncertainty are used to ensure the safe and ethical use of these novel tools. For machine learning models aimed for clinical application, uncertainty estimates may help improve model trustworthiness, guard against domain shift, and flag highly uncertain samples for manual expert review. Here, we provide a thorough assessment of deep learning patient-level uncertainty for histopathological diagnosis in a cancer application and its potential generalizability. We describe an algorithm for the separation of predictions into low- and high-confidence via uncertainty thresholding and show that high-confidence predictions have superior performance to low-confidence predictions in cross-validation, with balanced and unbalanced data, and with external evaluation. UQ thresholding remains a robust strategy when tested on data from multiple separate institutions, and even in the presence of domain shift when tested on cancers of a different primary site than the training data. This

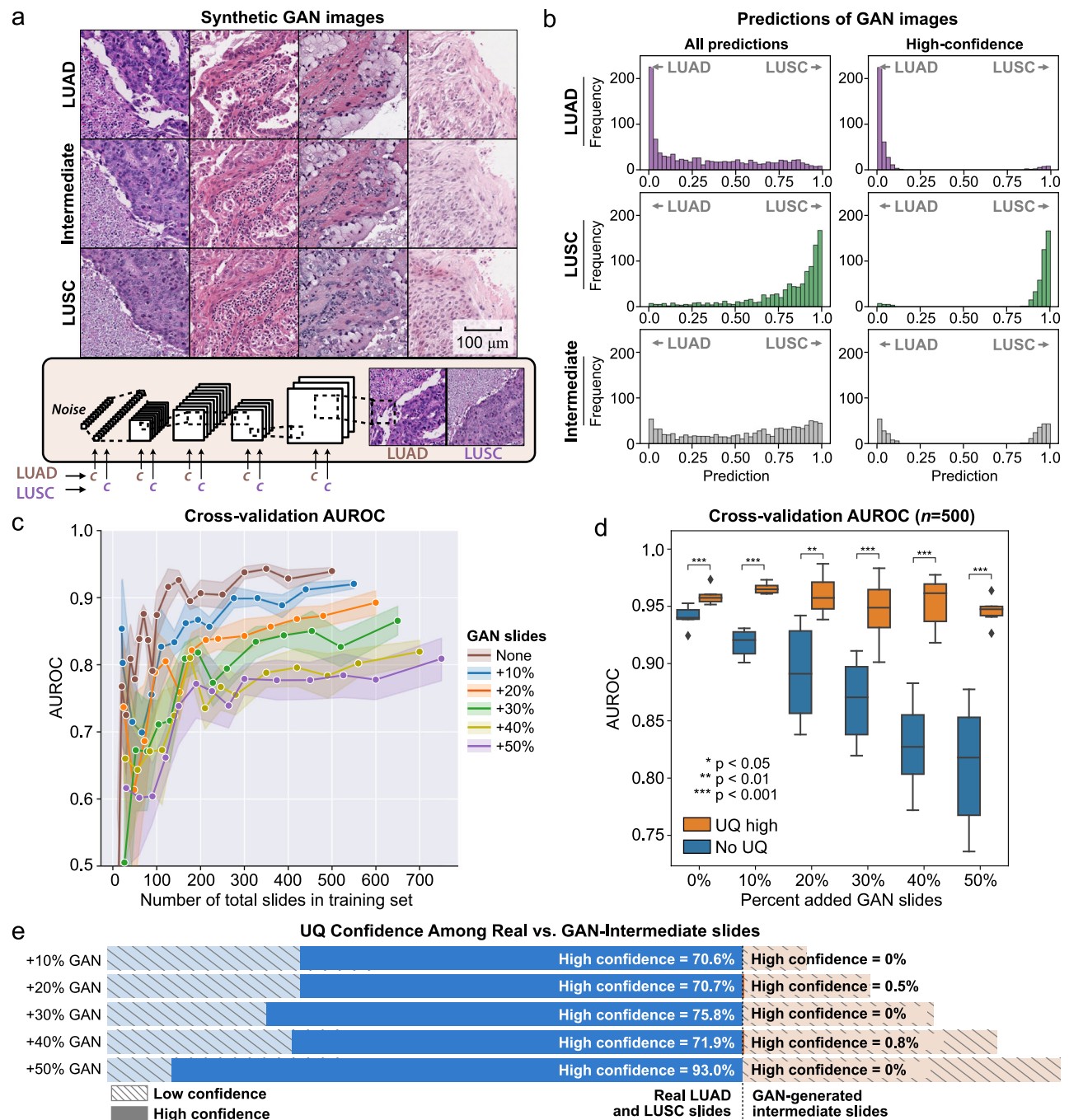

**Fig. 7 | Uncertainty thresholding in a synthetic test using GAN-generated images. a** A class-conditional generative adversarial network (GAN) was trained on TCGA to generate adenocarcinoma (LUAD) or squamous cell carcinoma (LUSC) synthetic images. Using embedding interpolation, intermediate neutral images are also generated to approximate images near the decision boundary. Example synthetic images are shown here using the LUAD, LUSC, and Intermediate class labels. **b** Using a model trained on the full TCGA dataset, predictions were calculated for 1000 LUAD, LUSC, and Intermediate GAN images. Synthetic LUAD and LUSC images were predicted accurately, and Intermediate synthetic images showed an even spread of predictions. **c** Models were trained with the addition of varying amounts of GAN-Intermediate slides with randomly assigned labels. Cross-validation slide-level area under receiver operator curve (AUROC) is shown. Performance degrades with increasing proportion of GAN-Intermediate slides. The shaded intervals represent the AUROC 95% confidence interval at each dataset size. **d** Cross-validation slide-level AUROC is shown from models trained with a dataset size of

500 slides plus varying amounts of GAN-Intermediate slides. Performance degrades as increasing number of uninformative GAN-Intermediate slides are added, but performance in the high-confidence uncertainty quantification (UQ) cohorts remains high despite large numbers of uninformative slides. For the +0%, +10%, +20%, +30%, +40%, and +50% GAN experiments, $p$ values comparing high-confidence AUROC to AUROC without UQ are 0.00032, 0.00014, 0.0034, <0.0001, 0.00068, and 0.00090, respectively. Statistical comparisons were made with one-sided, paired $t$-tests. **e** Distribution of low- and high-confidence predictions in the experiments shown in **d**. Virtually none of the GAN-Intermediate slides are classified as high-confidence in these experiments. For all boxplots, center line represents the median (50th percentile), lower and upper box bounds represent interquartile range (25th–75th percentile), and minimum (lower whisker) and maximum (upper whisker) bounds extend to furthest datapoint up to 1.5 times the interquartile range, with outliers shown as diamonds. Source data are provided as a Source Data file.

uncertainty thresholding paradigm excels at identifying decision-boundary uncertainty in a synthetic test using GANs, and expert pathologist review of low- and high-confidence predictions confirms the method's ability to select biologically unambiguous images.

Bayesian Neural Networks, which utilize dropout as a form of ensembling to approximate sampling of the Bayesian posterior, were among the first methods for uncertainty quantification in imaging-based convolutional neural networks[28]. The potential utility of BNNs and dropout-enabled networks to estimate uncertainty in histopathologic classification was explored by Syrykh et al, who utilized BNNs to differentiate the histopathological diagnosis of follicular lymphoma and follicular hyperplasia[33]. Their analysis revealed that predictions with high-confidence, as determined by a manually-chosen threshold, sustained high performance. Other groups have similarly investigated uncertainty estimation for histopathologic classification in breast cancer[34–37] and colorectal cancer[32,38].

Our work improves upon previous UQ analysis by providing rigorous, clinically-oriented performance assessment on external datasets of whole-slide images using thresholds determined on training data. We chose the histologic outcome of lung adenocarcinoma vs. squamous cell carcinoma to test our UQ methodology as it is a clinically relevant endpoint with occasional ambiguity in morphologic characteristics on standard hematoxylin and eosin (H&E) staining. Current International Association for the Study of Lung Cancer Pathology Committee (IASCLC) guidelines acknowledge the inherent histologic ambiguity that may exist in some tumors by recommending immunohistochemical (IHC) staining with p40 and TTF1 to differentiate between adenocarcinomas and squamous cell carcinomas[41]. Despite the ambiguity that may exist in some cases, however, Coudray et al. demonstrated feasibility of deep learning for classification of this outcome from standard H&E slides[7]. Their model suffered performance degradation when translated to an external dataset, however, and required supervised region-of-interest (ROI) annotation of slides by an expert pathologist. The described UQ thresholding strategy enables closer replication of deep learning model clinical application, with predictions generated on whole-slide images without requiring pathologist annotation. The uncertainty thresholding paradigm also enables higher accuracy, high-confidence predictions with external evaluation.

We have shown that high-confidence predictions consistently outperform low-confidence predictions across a spectrum of domain shifts when generated from models trained on datasets as small as 100 slides. Uncertainty thresholding enables consistently improved performance for high-confidence cases among different institutions, as tested with the multi-institution CPTAC cohort and a separate, single-institution dataset from Mayo Clinic. Furthermore, when model predictions were generated for an entirely separate distribution—cancers from non-lung primary sites—uncertainty thresholding yielded significant improvements in accuracy for the high-confidence predictions. Expert pathologist review of low- and high-confidence predictions confirms that high-confidence predictions are enriched for images with unambiguous morphology, supporting the biological relevance of the estimated uncertainty.

Actionable estimates of confidence and uncertainty will enable the development of safe and ethical models for clinical practice. Models designed for automating diagnosis, classification, or grading of tumors could use such a system to flag low-confidence predictions for additional testing or manual pathologist review, reducing errors while enabling trustworthy automation. Clinicians designing models to inform treatment response could opt to report and use only high-confidence predictions, decreasing the number of patients whose treatment is determined by a potentially erroneous prediction. Confidence estimates may also help with safe model deployment to new institutions and settings, where domain shift might otherwise compromise performance integrity.

While significant accuracy improvements are seen in high-confidence data, realizing these performance gains requires abstaining from predictions for a portion of the data. For the described application of lung cancer subtyping, approximately one-fourth of predictions are low-confidence, although the described algorithm may yield different proportions of high- and low-confidence predictions when applied to other domains and datasets. The proportion of high-confidence predictions for a given dataset will need to be determined empirically, and the maximum tolerated proportion of low-confidence predictions will be application specific.

Despite these promising results, several limitations of the above work must be acknowledged. Our extensive testing was performed on a binary classification model trained to a histologically well-defined outcome, conditions which will not be true for every application of deep learning for histopathology. Additionally, while the principles of uncertainty estimation extend to multi-class models and regression[29,42], further confirmatory testing of our thresholding strategy is needed for these applications. Although uncertainty thresholding flagged 78% of non-lung, non-adenocarcinoma, non-squamous OOD slides as low-confidence, new and emerging methods to improve robustness against OOD data, such as adversarial training[43], multi-head CNNs[44], and hyper-deep-ensembles[30], may help decrease the number of OOD slides erroneously reported with high-confidence. Finally, models trained on unbalanced datasets appear to benefit less from uncertainty thresholding, although it is unclear if this is due to the class imbalance or limitations in the number of slides available in the least-frequent outcome. In the largest 1:10 unbalanced experiment, for example, only 26 slides were available for training and uncertainty thresholding in the least-frequent outcome.

Herein we describe a fast, robust, and generalizable uncertainty thresholding algorithm to aid clinical deep neural networks tools for histopathology. We provide an automated method for high-confidence slide-level predictions, which increases accuracy in several real-world datasets. Uncertainty estimates are consistent with biological expectations when assessed by expert pathologists and robust against domain shift. This method is a significant step towards the practical implementation of uncertainty for clinical decision making with deep neural network-based tools for histopathology.

## Methods

This work complies with all ethical regulations as approved by the University of Chicago Institutional Review Board (IRB 20-0238). The data from Mayo Clinic was obtained with ethical approval by the Mayo Clinic Institutional Review Board (IRB 20-005623) and transferred between institutions with a Data Use Agreement.

### Dataset description

**Dataset sources.** Our training dataset contained 941 hematoxylin and eosin (H&E)-stained WSIs, comprised of 467 lung adenocarcinomas (LUAD) and 474 lung squamous cell carcinomas (LUSC) from The Cancer Genome Atlas (TCGA), with only one slide per patient[45]. Our first external evaluation dataset contained 1,306 slides (644 LUAD, 662 LUSC) spanning 416 patients (203 LUAD, 213 LUSC) from the multi-institution database Clinical Proteomic Tumor Analysis Consortium (CPTAC)[46]. Diagnoses were determined by the "primary_diagnosis" column in the TCGA clinical data and the "histologic_type" column in the CPTAC-LUAD data. Adenosquamous tumors were removed in all cohorts. No specific diagnosis column was available in the CPTAC squamous cell cohort (CPTAC-LSCC), so all slides were assigned the label "squamous cell". Our second evaluation set is a real-world, single-institution dataset from Mayo Clinic containing 190 slides (150 LUAD, 40 LUSC) spanning 186 patients (146 LUAD, 40 LUSC). Diagnosis of adenocarcinoma vs. squamous cell carcinoma for the Mayo Clinic cohort was rendered by histopathological review by an institutional pathologist. Patient characteristics for each dataset are shown in

Supplementary Table 1. For the TCGA dataset, a list of all diagnoses is provided in Supplementary Data 1, and a full case list with patient identifiers is provided in Supplementary Data 2. For the CPTAC dataset, a list of all diagnoses is provided in Supplementary Data 3, and a full case list with patient identifiers is provided in Supplementary Data 4. For the Mayo dataset, a list of all diagnoses is provided in Supplementary Data 5, and a full case list with patient identifiers is provided in Supplementary Data 6. Raw slide annotation files are provided for the TCGA dataset in Supplementary Data 7. Raw slide annotation files are provided for the CPTAC-LUAD and CPTAC-LSCC datasets in Supplementary Data 8, 9 respectively. Raw slide annotation files are provided for the Mayo dataset in Supplementary Data 10. Our out-of-distribution (OOD) experiment included 700 non-lung squamous cell cancers, 2456 non-lung adenocarcinomas, and 4015 non-lung, non-squamous, non-adenocarcinoma cancers. A full description of all pathologic diagnoses included in the OOD dataset is included in Supplementary Data 11.

**Image processing.** Image tiles were extracted from whole-slide images (WSI) with an image tile width of 302 µm and 299 pixels (effective magnification: 10X), using Slideflow[8,47]. Background image tiles were removed via grayspace filtering, Otsu's thresholding[48], and gaussian blur filtering. Gaussian blur filtering was performed with a sigma of 3 and threshold of 0.02. Experiments were performed on datasets with and without Otsu's thresholding and/or blur filtering and with varying grayspace fraction thresholds to confirm generalizability of the UQ methods regardless of background filtering method (Supplementary Figs. 6, 7). Image tiles underwent digital stain normalization using a modified Reinhard method, with brightness standardization disabled for computational efficiency[49]. For the lung TCGA training dataset only, image tiles were extracted only from within pathologist-annotated regions of interest (ROIs) to maximize cancer-specific training data. The median number of tiles per slide within the training dataset was 1026. During external evaluation, predictions are generated across all tiles from a given WSI. The median number of tiles per slide for the CPTAC dataset was 1181, with a median of 2299 tiles per slide for the Mayo dataset. When multiple slides were available for a given patient, patient-level predictions were made by aggregating all tiles from a patient's slides.

## Deep learning models

**Model architecture and hyperparameters.** We trained deep learning models using an Xception-based architecture with ImageNet pretrained weights and two hidden layers of width 1024, with dropout ($p = 0.1$) after each hidden layer. Models were trained with Slideflow[47] (version 1.1) using the Tensorflow backend with a single set of hyperparameters and category-level mini-batch balancing (Supplementary Table 2). Hyperparameters were chosen based on prior work[8] without further tuning in order to reduce the risk of overfitting on this dataset, with the exception of added dropout-enabled hidden layers and the use early stopping (enabled due to the large dataset size). Training data was augmented with random flipping/rotating, JPEG compression, and gaussian blur. We trained four pilot models at varying dataset sizes without early stopping to assess the optimal number of epochs and found the optimal number of epochs to be one, likely due to the amount of redundant morphologic information in a WSI (Supplementary Fig. 8). The remainder of our models were trained for one epoch with early stopping enabled. For non-UQ models, slide-level predictions are made by averaging the tile-level predictions for all image tiles from a given WSI, and in cases where a patient had multiple WSIs, patient-level predictions were made on tiles aggregated from a patient's slides.

## Estimation of uncertainty

Uncertainty is estimated with the Bayesian Neural Network (BNN) approach. This is an ensemble method where uncertainty is quantified as the disagreement of the predictions made by different models sampled from an ensemble of neural networks. The disagreement is computed as the standard deviation of the predictions by the sampled neural networks. BNN is a specific version of the ensemble method which differs from alternatives such as Deep Ensembles in the way that members of the ensemble are sampled: sampling is performed from a posterior distribution of models conditioned on the training data. Specifically, Gal and Ghahramani show that sampling from predictions generated via neural networks with Monte Carlo dropout is equivalent to sampling from a variational family (Gaussian Mixture), approximating the true deep Gaussian process posterior[28]. Thus, the distribution of predictions resulting from multiple forward passes in a dropout-enabled network approximates sampling of the Bayesian posterior of a deep Gaussian process, and the standard deviation of such a distribution is an estimate of predictive uncertainty[27,32,33,36–38,50,51].

During experiments with uncertainty quantification, an ensemble of models generates a distribution of predictions for each image tile with 30 forward passes in a dropout-enabled network (Fig. 1b). Let $y_j$ ($x_{\text{tile}}$) represent the prediction from a single model within the ensemble, which contains $j = 30$ models. The mean of all predictions in this distribution for a single image tile, $\hat{\mu}(x_{\text{tile}})$, is the final tile-level prediction. The standard deviation of this distribution, denoted as $\hat{\sigma}(x_{\text{tile}})$, is tile-level uncertainty.

$$U(x_{\text{tile}}) = \hat{\sigma}(x_{\text{tile}}) = \sqrt{\frac{\sum_{j=1}^{30}\left[y_j(x_{\text{tile}}) - \hat{\mu}(x_{\text{tile}})\right]^2}{30}} \tag{1}$$

We determine the optimal uncertainty threshold value, $\theta_{\text{tile}}$, below which image tiles are more likely to be correct, as compared to image tiles with higher uncertainty. To find the tile-level uncertainty threshold that optimally separates predictions into likely-correct (high-confidence) and likely-incorrect (low-confidence), we calculate the sensitivity and specificity for misprediction for all possible tile-level uncertainty thresholds $t_{\text{tile}}$. The corresponding Youden's index ($J$) for each uncertainty threshold $t_{\text{tile}}$ is then calculated as

$$J(t_{\text{tile}}) = Se(t_{\text{tile}}) + Sp(t_{\text{tile}}) - 1 \tag{2}$$

The optimal tile-level uncertainty threshold $\theta_{\text{tile}}$ is then defined as the threshold which maximizes the Youden index:

$$\theta_{\text{tile}} = \underset{t_{\text{tile}}}{\arg\max} J(t_{\text{tile}}) \tag{3}$$

This single threshold is then used for all predictions made by the model. We take a binary approach to confidence using the uncertainty threshold, with confidence of the image tile defined as

$$C(x_{\text{tile}}) = \begin{cases} \text{high-confidence} & \hat{\sigma}_{\text{tile}} < \theta_{\text{tile}} \\ \text{low-confidence} & \hat{\sigma}_{\text{tile}} \geq \theta_{\text{tile}} \end{cases} \tag{4}$$

Slide-level uncertainty is calculated as the mean of tile-level uncertainties for all high-confidence tiles from a slide. With $i$ representing the index of a given tile in a slide, slide-level uncertainty is defined as

$$U(x_{\text{slide}}) = \hat{\sigma}(x_{\text{slide}}) = \frac{\sum_{C(x_{\text{tile},i}) = \text{high-confidence}} U(x_{\text{tile},i})}{\sum_{C(x_{\text{tile},i}) = \text{high-confidence}} 1} \tag{5}$$

Similarly, for models using UQ, slide-level predictions $\hat{\mu}(x_{\text{slide}})$ are calculated by averaging high-confidence tile predictions:

$$\hat{\mu}(x_{\text{slide}}) = \frac{\sum_{C(x_{\text{tile},i}) = \text{high-confidence}} \hat{\mu}(x_{\text{tile},i})}{\sum_{C(x_{\text{tile},i}) = \text{high-confidence}} 1} \qquad (6)$$

As with tile-level uncertainty, the slide-level uncertainty threshold $\theta_{\text{slide}}$ is then determined by maximizing the Youden index ($J$) when slide-level uncertainty is formulated as a test for slide-level misprediction. Let $t_{\text{slide}}$ indicate a given slide-level uncertainty threshold. The optimal slide-level uncertainty threshold is defined as:

$$\theta_{\text{slide}} = \underset{t_{\text{slide}}}{\arg\max}\, J(t_{\text{slide}}) \qquad (7)$$

This single slide-level uncertainty threshold is used for all predictions made by the model. Slide-level confidence is then defined as

$$C(x_{\text{slide}}) = \begin{cases} \text{high-confidence} & \hat{\sigma}_{\text{slide}} < \theta_{\text{slide}} \\ \text{low-confidence} & \hat{\sigma}_{\text{slide}} \geq \theta_{\text{slide}} \end{cases} \qquad (8)$$

Finally, the slide-level prediction threshold is determined by maximizing the Youden index ($J$) on a validation dataset to optimize correct predictions. This threshold is then used for all external evaluation dataset testing for a given model. Let $t_{\text{pred}}$ indicate a slide-level prediction threshold. The optimal threshold is formally defined as

$$\theta_{\text{pred}} = \arg\max J(t_{\text{pred}}) \qquad (9)$$

## Nested cross-validation for uncertainty thresholds

Optimal $\theta_{\text{tile}}$ and $\theta_{\text{slide}}$ should not be determined from validation data, as these thresholds require knowledge of correct labels, and use of these labels would constitute a form of data leakage. Thus, we use a nested cross-validation strategy in which optimal thresholds are determined from within training data only, and then applied to validation data (Fig. 1d). Within a given outer cross-fold training dataset, data is segmented into five nested cross-folds, and optimal $\theta_{\text{tile}}$ and $\theta_{\text{slide}}$ are determined for each of the five inner cross-fold validation datasets. The final $\theta_{\text{tile}}$ is defined as the minimum $\theta_{\text{tile}}$ across each of the nested cross-fold values, and $\theta_{\text{slide}}$ is defined as the maximum $\theta_{\text{slide}}$ across the nested cross-folds. These thresholds are then used for separating validation data into low- and high-confidence. This nested training strategy mitigates the data leak at the cost of requiring larger training dataset sizes.

## Statistics & Reproducibility

**Cross-validation without UQ.** To test the effect of increasing dataset size on cross-validated performance, we trained models using progressively increasing amounts of data, beginning with a sample size of only 10 slides and increasing to the maximum dataset size of 941 slides. For each sample size, we bootstrapped three-fold cross-validation four times, for a total of 12 models trained per dataset size. Across a total of 23 tested dataset sizes, this yielded 276 total models. Mean Area Under Receiver Operator Curves (AUROC) are reported as mean ± SEM.

**Cross-validation with uncertainty thresholding.** For dataset sizes greater than 100 slides, we bootstrapped three-fold cross-validation twice using a dropout-enabled network, generating both tile- and slide-level predictions and uncertainties for validation data. No statistical method was used to predetermine this sample size. Validation data thresholding into low- and high-confidence is then performed as described above using nested 5-fold cross-validation within training data. This strategy resulted in a total of 36 models trained at each dataset size; across 14 dataset sizes, this yielded a total of 504 models.

For each dataset size, we compare the distribution of non-UQ validation AUROCs to the high-confidence UQ cohort AUROCs, with statistical comparison performed using one-sided, paired $t$-tests. No data were excluded from the analyses.

**Cross-validation with unbalanced outcomes.** To investigate the effect of unbalanced data on both cross-validation performance and utility of UQ, we also bootstrapped training of three-fold cross-validation twice with 1:3/3:1 and 1:10/10:1 ratio of LUAD:LUSC, for a total of 6 models trained at each dataset size of 50, 100, 150, 200, 300, and 400 slides. No statistical method was used to predetermine this sample size. AUROCs at each dataset size were plotted with box-plots. Box-plots here and in all areas of this manuscript are shown with center line representing median, box limits representing upper and lower quartiles, and whiskers representing 1.5× interquartile range, and outliers represented as diamond points. No data were excluded from the analyses.

**Full model training and threshold determination.** For each tested sample size, a single model was trained with early stopping using the average early-stopped batch from cross-validation. The constituent cross-validation models were also used to determine the optimal slide-level prediction threshold $\theta_{\text{pred}}$ to be applied on the external datasets, as well as the UQ thresholds, if applicable (Fig. 1c).

**Activation maps.** To investigate the uncertainty landscape on an external dataset, a single UQ model trained on the full TCGA dataset ($n = 941$) was used to calculate penultimate layer activations, predictions, and uncertainty for 10 randomly selected image tiles from each slide in the CPTAC dataset. Activations were plotted with UMAP[52] for dimensionality reduction, and corresponding tile images were overlaid onto the plot in a grid-wise fashion to create a mosaic map[8]. Images features at different locations of the mosaic map were reviewed with two expert pathologists. Corresponding UMAP plots were labeled with tile-level predictions, classification prediction, uncertainty, and confidence via thresholding.

**Out-of-distribution testing on other cancer types.** To test our uncertainty thresholding strategy on OOD data, we generated predictions for 7171 WSIs from 28 different non-lung cancer cohorts (broadly separated into squamous, adenocarcinoma, and other), using the UQ model trained on the full TCGA dataset (941 slides). Uncertainty thresholding was performed and predictions from the high-confidence cohort are displayed. No data were excluded from the analyses.

## Synthetic testing with GANs

To test the ability of UQ to identify and filter out non-informative images as low-confidence, we designed a synthetic test using a class-conditional generative adversarial network (GAN). We trained StyleGAN2 on the TCGA training dataset to generate synthetic image tiles using the LUAD and LUSC class labels[53]. Using latent space embedding interpolation, we generate morphologically neutral images near the LUAD/LUSC decision boundary, which we designate GAN-Intermediate. We created GAN-intermediate synthetic slides by aggregating 1000 random GAN-Intermediate tile images together and tested the effect of adding varying amounts of these morphologically neutral GAN-Intermediate slides to our cross-validation dataset. These GAN-intermediate slides are labeled with a random (LUSC vs LUAD) diagnosis. We then test the ability of the described UQ algorithm to discard the GAN-Intermediate slides as low-confidence by training models in cross-validation with varying percentages of GAN-Intermediate slides added to both the training and validation datasets. This method approximates the nontrivial but difficult to quantify presence of ambiguous images in real-world datasets, allowing us to titrate the amount of informative data available in both training and

validation. This experiment with StyleGAN2 was only performed once due to computational restraints.

## Reporting summary

Further information on research design is available in the Nature Research Reporting Summary linked to this article.

## Data availability

Whole-slide images from the The Cancer Genome Atlas (TCGA) lung adenocarcinoma project are publicly available at lung adenocarcinoma project [https://portal.gdc.cancer.gov/projects/TCGA-LUSC] ", and images from the lung squamous cell project are available at " lung squamous cell project [https://portal.gdc.cancer.gov/projects/TCGA-LUAD] ". Whole-slide images from the Clinical Proteomic Tumor Analysis Consortium (CPTAC) lung adenocarcinoma and lung squamous cell carcinoma collections are available at "CPTAC [https://www.cancerimagingarchive.net/collections/] ", under the collection names "CPTAC-LUAD" and "CPTAC-LSCC", respectively. A manifest of TCGA and CPTAC slide identifiers for images used in this study is included in the Supplementary Data. Restrictions apply to the availability of the internal Mayo Clinic dataset, which was used with institutional permission for the current study and is thus not publicly available due to data privacy laws. All requests will be promptly evaluated based on institutional and departmental policies to determine whether the data requested are subject to intellectual property or patient privacy obligations. Data can only be shared for non-commercial academic purposes and will require a data user agreement. All other relevant data supporting the key findings of this study are available within the article and its Supplementary Information files or from the corresponding author upon reasonable request. Source data are provided with this paper.

## Code availability

All code necessary to reproduce the results of this manuscript, including the uncertainty thresholding algorithm, are provided at https://github.com/jamesdolezal/biscuit and deposited to Zenodo[54]. Deep learning analyses were performed using Slideflow 1.1.0 (https://github.com/jamesdolezal/slideflow) with Tensorflow 2.7.1 as the primary deep learning package. This manuscript utilized QuPath 0.3.0 for region-of-interest pathologist annotations.

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

## Acknowledgements

This work was funded by the National Institute of Health/National Cancer Institute NIH/NCI) U01-CA243075 (A.T.P.), National Institute of Health/National Institute of Dental and Craniofacial Research (NIH/NIDCR) R56-DE030958 (A.T.P.), NIH/NCI R21 CA251923 (A.M.), Department of Defense W81XWH-22-1-0021 Concept Award (A.M.), Mark Foundation ASPIRE Award (A.M.), grants from Cancer Research Foundation (A.T.P.), grants from Stand Up to Cancer (SU2C) Fanconi Anemia Research Fund – Farrah Fawcett Foundation Head and Neck Cancer Research Team Grant (A.T.P.), Horizon 2021-SC1-BHC I3LUNG grant (A.T.P and M.C.C.), and grants from ECOG Research and Education Foundation (A.S.).

## Author contributions

J.M.D. conceived the study. J.M.D., A.S., D.K., and A.T.P. designed the experiments and the uncertainty thresholding algorithm. J.M.D. wrote the software and performed the experiments. J.M.D., A.S., D.K., Siddhi Ramesh, S.K., and A.T.P. analyzed data. B.C. and A.N.H. performed histologic analysis and pathologist review. J.M.D., A.S., and D.K. wrote the manuscript. A.S.M., Sagar Rakshit, R.B., M.C.B., A.O.B., and J.J.S. acquired and processed the dataset from Mayo Clinic. E.E.V. and M.C.G. provided clinical insight, and all authors contributed to general discussion and revision of the manuscript.

## Competing interests

D.K. is an employee of DV Group, LLC, but holds no competing interests for this work. A.T.P. reports no competing interests for this work, and reports personal fees from Prelude Therapeutics Advisory Board, Elevar Advisory Board, AbbVie consulting, Ayala Advisory Board, and stock options ownership in Privo Therapeutics, all outside of submitted work. E.E.V. is a consultant and provides advisory roles for AstraZeneca, Beigene, BioNTech, Eli Lilly, EMD Serono, Genentech/Roche, GlaxoSmithKline, and Novartis, with no competing interests for this work. All remaining authors report no competing interests.
