## [Peer Review File · Nature Communications]

REVIEWER COMMENTS

Reviewer #1 (Remarks to the Author):

In this work, the authors developed a method to determine the cutoffs for low- and high-confidence predictions for uncertainty quantification in digital histopathology. Using the TCGA data, they compared many different UQ-enabled models with non-UQ models for classifying lung squamous cell carcinoma and lung adenocarcinoma. They concluded that the high-confidence predictions from UQ models outperform non-UQ models in terms of classification accuracy. I think the authors addressed an important question but the proposed method was not well described. Also there were lack of systematic comparisons with existing UQ methods. I have the following specific comments.

1. On Page 3, lines 92 - 93, the authors stated “we describe a clinically-oriented method for determining slide-level confidence using the Bayesian approach to estimating uncertainty, with uncertainty thresholds determined from training data”. I don’t understand why the proposed method is a “Bayesian approach”. The Section “Estimation of Uncertainty” from lines 425 - 467 is quite confusing to me. A lot of notations are not well defined. For a Bayesian approach, what is the likelihood and what is the prior. How to compute the posterior? All those details are missing.
2. It would be helpful if the authors explain why the proposed method particularly works well for digital image application. The method does not incorporate any the unique features of image applications in the procedure of uncertainty estimation.
3. There are a lot of tuning parameters In the UQ-enable and non-UQ deep learning models, e.g., the dropout rate, the network architectures. It would be helpful if the authors provide some justifications on the choices of those parameters.
4. The imaging processing procedure may have large impact on the classification accuracy and uncertainty quantifications. In particular, the steps for removing background image tiles including grayscale filtering, Otsu’s thresholding and gaussian blur detection. The authors should provide more details on those procedures and show the sensitivity of the UQ to the mild changes of tuning parameters in those procedures.

Reviewer #2 (Remarks to the Author):

This paper presented an uncertainty quantification (UQ) algorithm to aid clinical decision using deep neural network for digital histopathology. The method estimated uncertainty using dropout and calculated thresholds using training data to define low-confidence and high-confidence predictions. In empirical studies, the authors trained models to distinguish lung adenocarcinoma vs. squamous cell

carcinoma, and obtained the following promising results: 1) uncertainty thresholding improved accuracy for high-confidence predictions; 2) uncertainty thresholding could generalize to out-of-distribution data; 3) areas of high uncertainty correlated with histologic ambiguity; and 4) UQ thresholding identified decision-boundary uncertainty. The paper is well-structured and clearly written. The methods are technically sound, and the results are promising. The following are a few comments that I hope can help further improve the manuscript.

- “A total of 276 standard (non-UQ) and 504 UQ-enabled DCNN models based on the Xception architecture were trained to discriminate between lung squamous cell carcinoma and lung adenocarcinoma using varying amounts of data from TCGA” It is unclear why different numbers of non-UQ (276) and UQ-enabled (504) DCNN models were trained here. It would be helpful if the authors could provide some relevant discussion on how to choose these two specific numbers (i.e., 276 vs 504).
- In Figure 2b, it would be helpful to show also the percentage of the UQ high-confidence predictions over all the predictions.
- “Figure 2(c) Across all cross-validation experiments with UQ, a median of 84.6% (43.8% - 100%) of validation data is classified as high-confidence. The shaded interval represents the 95% confidence interval at each dataset size.” The authors may also want to report the mean percentage in addition to the median percentage.
- “Figure 3(a) Models trained on TCGA at varying dataset sizes were validated on lung adenocarcinomas and squamous cell carcinomas from CPTAC. Patient-level metrics are shown with the dotted lines, and slide-level metrics are shown with Xs. AUROC, accuracy, and Youden’s J are all improved in the high-confidence UQ cohorts. The proportion of patients and slides reported as high-confidence is shown in the last panel.” The slide-level metrics are shown with Xs, which is a bit hard to read.
- It would be helpful to include more discussion on low confidence predictions. For these predictions, one could not make any decision. Readers might be interested in learning how many are these low confidence cases. If there are too many, the proposed methods may not be that effective in practical applications.

Response to Reviewers

Reviewer #1

In this work, the authors developed a method to determine the cutoffs for low- and high-confidence predictions for uncertainty quantification in digital histopathology. Using the TCGA data, they compared many different UQ-enabled models with non-UQ models for classifying lung squamous cell carcinoma and lung adenocarcinoma. They concluded that the high-confidence predictions from UQ models outperform non-UQ models in terms of classification accuracy. I think the authors addressed an important question but the proposed method was not well described. Also there were lack of systematic comparisons with existing UQ methods. I have the following specific comments.

1. On Page 3, lines 92 - 93, the authors stated “we describe a clinically-oriented method for determining slide-level confidence using the Bayesian approach to estimating uncertainty, with uncertainty thresholds determined from training data”. I don’t understand why the proposed method is a “Bayesian approach”. The Section “Estimation of Uncertainty” from lines 425 - 467 is quite confusing to me. A lot of notations are not well defined. For a Bayesian approach, what is the likelihood and what is the prior. How to compute the posterior? All those details are missing.

Thank you for raising this question. The use of the term “Bayesian approach” comes from prior work (particularly Gal et al) which established that the distribution from a neural network obtained via dropout approximates sampling of the Bayesian posterior of a deep Gaussian process, and that standard deviation of such a distribution expresses the model’s predictive uncertainty. This work forms the foundation of our uncertainty estimation paradigm, which provides a novel method of aggregating individual image tile uncertainty into whole-slide uncertainty and confidence thresholding. We will highlight to readers the paper by Gal et al, *Dropout as a Bayesian Approximation: Representing Model Uncertainty in Deep Learning*, for proofs and mathematical justification for this approach.

We appreciate that there are many alternative methods for estimating uncertainty for individual image tiles, including test-time augmentation, deep ensembles, and hyper-deep ensembles, methods which have previously been compared by other groups (Thagaard, Poceviciute). We chose one well-studied method for estimating tile-level uncertainty, MC Dropout, in order to describe a novel method of aggregating tile-level uncertainty into patient-level uncertainty and confidence.

We have significantly rewritten the section “Estimation of Uncertainty” in the Methods to better explain the foundational methodology used for our uncertainty estimation and improved notation clarity. We also included a new paragraph in this section.

Changes:

Updates to Figure 1, with new panels **d** and **e**.

Figure 1. Estimation of uncertainty and confidence thresholding. (a) With standard deep learning neural network designs, a single image yields a single output prediction. When dropout is enabled during inference, predictions for a single image will vary based on which nodes are randomly dropped out. To estimate tile-level uncertainty, images first undergo 30 forward passes through the network, resulting in a distribution of predictions. The mean of each prediction, $\hat{\mu}(x_{tile})$, represents the final tile-level prediction, and the standard deviation, $\hat{\sigma}(x_{tile})$, represents the tile-level uncertainty. (b) When UQ methods are used, incorrect predictions are associated with higher uncertainties than correct predictions³²⁻³⁸. From a given distribution of tile- or slide-level uncertainties, we determine the uncertainty threshold θ which optimally separates correct and incorrect predictions by maximizing Youden's index (J). Predictions with uncertainty below this threshold are high-confidence, and all others are low-confidence. (c) To prevent data leakage and overfitting, optimal tile- and slide-level uncertainty thresholds are determined through nested cross-validation within training folds. (d) Schematic for calculating tile-level uncertainty and confidence. The optimal tile-level uncertainty threshold θ_{tile} is calculated from a given validation dataset. Tiles from the dataset are separated into high- and low-confidence by whether the tile-level uncertainty falls below or above θ_{tile} , respectively. (e) Schematic for slide-level uncertainty and confidence. Slide-level uncertainty is defined as the average uncertainty among high-confidence tiles for a given slide. The optimal slide-level uncertainty threshold θ_{slide} is found and used to classify slides as high- and low-confidence.

Changes in paragraph text and equations in the Method's section "Estimation of Uncertainty", including the addition of a new paragraph:

“Uncertainty is estimated with the Bayesian Neural Network (BNN) approach. This is an *ensemble* method where the uncertainty is quantified as the “disagreement” of the predictions made by different models sampled from an ensemble of neural networks. All of the networks in the ensemble explain the same training data but can disagree on some images. The disagreement is computed simply as the standard deviation of the predictions by the sampled neural networks. BNN is a specific version of the ensemble method which differs from alternatives such as Deep Ensembles in the way that members of the ensemble are sampled: sampling is performed from a posterior distribution of models conditioned on the training data. Specifically, Gal and Ghahramani show that sampling from predictions generated via neural networks with Monte Carlo dropout is equivalent to sampling from a variational family (Gaussian Mixture), approximating the true deep Gaussian process posterior²⁷. Thus, the distribution of predictions resulting from multiple forward passes in a dropout-enabled network approximates sampling of the Bayesian posterior of a deep Gaussian process, and the standard deviation of such a distribution is an estimate of predictive uncertainty^{28, 50, 32-36, 51}.”

References:

27. Gal, Y. and Z. Ghahramani Dropout as a Bayesian Approximation: Representing Model Uncertainty in Deep Learning, PMLR. **48**: 1050-1059.
35. Thagaard, J., Hauberg, S., van der Vegt, B., Ebstrup, T., Hansen, J.D., Dahl, A.B. (2020). Can You Trust Predictive Uncertainty Under Real Dataset Shifts in Digital Pathology?. In: , *et al.* Medical Image Computing and Computer Assisted Intervention – MICCAI 2020. MICCAI 2020. Lecture Notes in Computer Science(), vol 12261. Springer, Cham. https://doi.org/10.1007/978-3-030-59710-8_80
36. Pocevičiūtė, M., Eilertsen, G., Jarkman, S. *et al.* Generalisation effects of predictive uncertainty estimation in deep learning for digital pathology. *Sci Rep* **12**, 8329 (2022). <https://doi.org/10.1038/s41598-022-11826-0>

2. It would be helpful if the authors explain why the proposed method particularly works well for digital image application. The method does not incorporate any the unique features of image applications in the procedure of uncertainty estimation.

We appreciate the reviewer’s concern. We did not mean to imply that the proposed method is specifically high performing in the imaging domain. We hoped to emphasize that methodologies such as these are not routinely implemented in imaging-based deep learning research, despite the importance of uncertainty estimation for clinical implementation where model predictions may be used to make high-risk decisions, for example relating to selecting treatment for a patient. Deep learning model interpretability and uncertainty are particularly challenging for imaging data, where there is no tangible relationship between input data points (pixels) and model output (predictions). MC dropout provides a robust and convenient method of estimating deep learning model uncertainty which works well for imaging data but has also been used for many other types of high-dimensional input modalities.

3. There are a lot of tuning parameters in the UQ-enabled and non-UQ deep learning models, e.g., the dropout rate, the network architectures. It would be helpful if the authors provide some justifications on the choices of those parameters.

Thank you for the opportunity to expand upon our hyperparameter selection process.

Our objective in this work was to develop and rigorously assess a novel uncertainty quantification approach for whole-slide imaging, rather than attempting to develop the best possible performing model for predicting lung adenocarcinoma vs. squamous cell carcinoma. To avoid the risk of overfitting

on this dataset, we used the same hyperparameters as previously published (Dolezal, 2021), with the exception of hidden layers with dropout (used for uncertainty estimation) and early stopping (used to reduce potential overfitting on this very large training dataset).

There is no standard consensus for the optimal dropout rate for uncertainty estimation for medical imaging applications; prior work has used dropout rates of 0.2, 0.5, and 0.6 (Srykh, Raczowski, Thagaard, Poceviute, Ponzio, Leibig, Song). Dropout is a regularization technique which can help with generalizability and overfitting, but as with other regularization methods, too strong of regularization can worsen accuracy (Kamalov, 2020). We chose a low dropout rate of 0.1 to decrease the likelihood of performance degradation when the method is applied to new datasets and problems. As with other hyperparameters, optimal dropout rate will be dataset-dependent and should be experimentally tuned if maximum performance is desired.

We recognize that other readers may also have questions about our hyperparameter selection, and have thus included additional information in the Methods regarding how these hyperparameters were chosen.

References:

- F. Kamalov and H. H. Leung, "Deep learning regularization in imbalanced data," 2020 *International Conference on Communications, Computing, Cybersecurity, and Informatics (CCCI)*, 2020, pp. 1-5, doi: 10.1109/CCCI49893.2020.9256674.

Changes (in Methods, under “Model architecture and hyperparameters”):

“... Hyperparameters were chosen based on prior work⁸ without further tuning in order to reduce the risk of overfitting on this dataset, with the exception of added dropout-enabled hidden layers and the use early stopping (enabled due to the large dataset size).”

4. The imaging processing procedure may have large impact on the classification accuracy and uncertainty quantifications. In particular, the steps for removing background image tiles including grayscale filtering, Otsu’s thresholding and gaussian blur detection. The authors should provide more details on those procedures and show the sensitivity of the UQ to the mild changes of tuning parameters in those procedures.

We appreciate the opportunity to expand upon the impact of slide-level image processing on classification accuracy and uncertainty estimation. We have added a reference for the Otsu’s thresholding algorithm and additional details regarding Gaussian blur filtering in the Methods section.

Our background and artifact filtering process includes two slide-level steps and one tile-level step. Gaussian blur filtering and Otsu’s thresholding are both slide-level steps which identify areas of background or artifact on whole-slide images; tiles are not extracted from these areas of the slide. The final tile-level step is grayscale filtering, where image tiles are converted to the HSV colorspace, pixels are thresholded into areas of color (foreground) and grayscale (background) based on saturation, and then tiles are discarded if the total grayscale fraction for the image exceeds some threshold.

To test the impact of the slide-level image processing steps on classification accuracy and uncertainty quantification, we extracted tiles from the training dataset with blur filtering and Otsu's thresholding, blur filtering alone, Otsu's thresholding alone, and no slide-level background filtering. We then trained models in cross-validation at the maximum dataset size a total of four times for each method (12 total models trained for each method), and then performed nested uncertainty estimation for each model to determine UQ thresholds (**Supplementary Figure 6**). High-confidence predictions outperformed predictions without UQ estimation for all slide processing methods ($P < 0.001$).

To investigate the potential impact of the grayscale filtering threshold on uncertainty quantification, we extracted all image tiles, without background filtering, for 50 adenocarcinomas and 50 squamous cell carcinomas in the CPTAC database. For each image tile, we calculated grayscale fraction, UQ-enabled model prediction, and estimated uncertainty. We plotted density estimation for image tiles at each grayscale value, separated by whether the prediction was correct or incorrect (**Supplementary Figure 7, A**). These results show a bimodal distribution for grayscale fraction, with a large peak around 0 (representing image tiles with little background) and another peak around 1 (indicating tiles mostly background). Image tiles with low grayscale fraction are much more likely to be correctly predicted than incorrectly predicted, whereas image tiles with grayscale fraction above 0.8 are just as likely to be correct as incorrect. We then plotted density estimation for grayscale fraction vs. uncertainty for each image tile, separated by whether the model prediction was correct or incorrect (**Supplementary Figure 7, B and C**). These results show that when grayscale fraction is low (less than 0.2), most correct predictions are below the uncertainty threshold, while most incorrect predictions are above the uncertainty threshold (and would thus be filtered out). When grayscale fraction exceeds around 0.8, there is an increase in the number of incorrectly predicted image tiles that fall below the uncertainty threshold and would fail to be removed by UQ thresholding. These results support a grayscale fraction threshold of around 0.7 – 0.8 to maximize the utility of uncertainty estimation to enrich for correct predictions.

We have included these results in the Supplementary Information, as shown below.

Changes to Methods (under “Image processing”):

“Background image tiles were removed via grayscale filtering, Otsu's thresholding⁴⁸, and gaussian blur filtering. Gaussian blur filtering was performed with a sigma of 3 and threshold of 0.02. Experiments were performed on datasets with and without Otsu's thresholding and/or blur filtering and with varying grayscale fraction thresholds to confirm generalizability of the UQ methods regardless of background filtering method (**Supplementary Figs. 6 and 7**).”

Changes to Supplementary Information:

Supplementary Fig. 6. Uncertainty thresholding improves predictions regardless of slide-level background processing. Models were trained in three-fold cross-validation, bootstrapped four times, on the full TCGA training dataset with varying slide-level background and artifact filtering methods. We trained models either with Gaussian blur filtering and Otsu’s thresholding, only Otsu’s thresholding, only Gaussian blur filtering, or no background filtering method (only tile-level grayscale filtering). For each model, we trained models in five-fold nested cross-validation to determine uncertainty thresholds. In all cases, high-confidence UQ predictions as determined by thresholds from nested cross-validation outperformed predictions from models without UQ ($p < 0.001$ in all cases).

Supplementary Fig. 7. Assessment of the interaction between grayscale fraction and uncertainty. To investigate the potential impact of grayscale filtering on uncertainty quantification, we extracted all image tiles, without background filtering, from 50 lung adenocarcinomas and 50 lung squamous cell carcinomas in the CPTAC dataset. For each image tile, we calculated grayscale fraction, UQ-enabled model prediction, and estimated uncertainty. **(a)** Kernel density estimation for image tiles with varying grayscale fractions, separated by whether the prediction was correct or incorrect. There is a bimodal distribution of grayscale fraction in this dataset. Image tiles with low grayscale fraction (< 0.2) are more likely to be correctly predicted, and image tiles with high grayscale fraction (> 0.8) are just as likely to be correct as

incorrect. (b) Two-dimensional kernel density estimation of grayscale fraction vs. uncertainty estimation for correctly predicted image tiles. When grayscale fraction is low, most correctly predicted image tiles fall below the uncertainty threshold and are thus classified as high-confidence. When grayscale fraction is high, most correct predictions fall above the uncertainty threshold and are thus filtered out as low-confidence. (c) Two-dimensional kernel density estimation of grayscale fraction vs. uncertainty estimation for incorrectly predicted image tiles. With high grayscale fraction, there is an increase in the number of incorrect predictions falling below the uncertainty threshold (erroneously classified as high-confidence) compared to correct predictions. These results support a grayscale fraction threshold of around 0.7 – 0.8 to maximize the utility of uncertainty estimation to enrich for correct predictions.

Reviewer #2 (Remarks to the Author):

This paper presented an uncertainty quantification (UQ) algorithm to aid clinical decision using deep neural network for digital histopathology. The method estimated uncertainty using dropout and calculated thresholds using training data to define low-confidence and high-confidence predictions. In empirical studies, the authors trained models to distinguish lung adenocarcinoma vs. squamous cell carcinoma, and obtained the following promising results: 1) uncertainty thresholding improved accuracy for high-confidence predictions; 2) uncertainty thresholding could generalize to out-of-distribution data; 3) areas of high uncertainty correlated with histologic ambiguity; and 4) UQ thresholding identified decision-boundary uncertainty. The paper is well-structured and clearly written. The methods are technically sound, and the results are promising. The following are a few comments that I hope can help further improve the manuscript.

1. “A total of 276 standard (non-UQ) and 504 UQ-enabled DCNN models based on the Xception architecture were trained to discriminate between lung squamous cell carcinoma and lung adenocarcinoma using varying amounts of data from TCGA” It is unclear why different numbers of non-UQ (276) and UQ-enabled (504) DCNN models were trained here. It would be helpful if the authors could provide some relevant discussion on how to choose these two specific numbers (i.e., 276 vs 504).

Thank you for the opportunity clarify how we calculated these numbers.

For non-UQ models, we describe in the methods section under “Training strategy” that models were trained with “bootstrapped three-fold cross-validation four times, for a total of 12 models trained per dataset size.” We bootstrapped our cross-validation experiments to increase statistical power. We tested a total of 23 dataset sizes, for a total of $23 \times 12 = 276$ total non-UQ models.

For the UQ models, we describe in this methods section that “for dataset sizes greater than 100 slides, we bootstrapped three-fold cross-validation twice using a dropout-enabled network.... validation data thresholding into low- and high-confidence is then performed using nested 5-fold cross-validation within training data”. We tested our UQ methodology on a total of 14 dataset sizes greater than or equal to 100 slides. At each dataset size, we bootstrapped three-fold cross-validation twice, for a total of 6 models trained. For each of these models, uncertainty thresholds were determined through nested 5-

fold cross-validation. Thus, the total number of models trained per dataset size is 6 models + (6 × 5 nested cross-fold models) = 36 models. Across 14 dataset sizes, this yields a total of 14 × 36 = 504 models for the UQ experiments.

We have clarified how these numbers were calculated by expanding the “Training strategy” section of the Methods:

Changes in Methods (changes are highlighted)

“Cross-validation without UQ. To test the effect of increasing dataset size on cross-validated performance, we trained models using progressively increasing amounts of data, beginning with a sample size of only 10 slides and increasing to the maximum dataset size of 941 slides. For each sample size, we bootstrapped three-fold cross-validation four times, for a total of 12 models trained per dataset size. Across a total of 23 tested dataset sizes, this yielded 276 total models. Mean Area Under Receiver Operator Curves (AUROC) are reported as mean ± SEM.”

“Cross-validation with uncertainty thresholding. For dataset sizes greater than 100 slides, we bootstrapped three-fold cross-validation twice using a dropout-enabled network, generating both tile- and slide-level predictions and uncertainties for validation data. Validation data thresholding into low- and high-confidence is then performed as described above using nested 5-fold cross-validation within training data. This strategy resulted in a total of 36 models trained at each dataset size; across 14 dataset sizes, this yielded a total of 504 models.”

2. In Figure 2b, it would be helpful to show also the percentage of the UQ high-confidence predictions over all the predictions.

Figure 2(c) Across all cross-validation experiments with UQ, a median of 84.6% (43.8% - 100%) of validation data is classified as high-confidence. The shaded interval represents the 95% confidence interval at each dataset size.” The authors may also want to report the mean percentage in addition to the median percentage.

Thank you for these suggestions. We expanded the figure legend for Figure 2 to include the total percentage of UQ high-confidence predictions across all experiments, which is 81.9%. We additionally included range of the mean percentage of high-confidence predictions at each dataset size (82.8%), as we did with the median percentage at each dataset size (84.6%).

Changes to Figure 2 Legend:

“Across all cross-validation experiments with UQ, 81.9% of predictions are classified as high-confidence. At each dataset size, the average percent of validation predictions classified as high-confidence is 82.8%, with a median of 84.6% (43.8% - 100%).”

3. “Figure 3(a) Models trained on TCGA at varying dataset sizes were validated on lung adenocarcinomas and squamous cell carcinomas from CPTAC. Patient-level metrics are shown

with the dotted lines, and slide-level metrics are shown with Xs. AUROC, accuracy, and Youden's J are all improved in the high-confidence UQ cohorts. The proportion of patients and slides reported as high-confidence is shown in the last panel." The slide-level metrics are shown with Xs, which is a bit hard to read.

Thank you for this feedback, we have improved readability of these graphs by making the X's more bold.

- It would be helpful to include more discussion on low confidence predictions. For these predictions, one could not make any decision. Readers might be interested in learning how many are these low confidence cases. If there are too many, the proposed methods may not be that effect in practical applications.

Thank you for the opportunity to expand upon the implications of low-confidence predictions. For the described application of lung cancer subtyping, one-fourth of predictions are low-confidence. The maximum tolerated proportion of low-confidence predictions will be application specific. For diagnostic applications in high-prevalence diseases with easily accessible second-line testing (such as manual pathologist review), a relatively high number of low-confidence cases may be tolerated if accurate predictions on a portion of the data improves pathologist workflow. For diagnostic applications in low-prevalence diseases without easily accessible pathologist review, a high number of low-confidence predictions may decrease the practicality of such a model.

We have included additional discussion regarding the implications of abstaining from low-confidence predictions in the Discussion, as included below.

New paragraph (in Discussion):

“While significant accuracy improvements are seen in high-confidence data, realizing these performance gains requires abstaining from predictions for a portion of the data. For the described application of lung cancer subtyping, approximately one-fourth of predictions are low-confidence, although the described algorithm may yield different proportions of high- and low-confidence predictions when applied to other domains and datasets. The proportion of high-confidence predictions for a given dataset will need to be determined empirically, and the maximum tolerated proportion of low-confidence predictions will be application specific.”

REVIEWERS' COMMENTS

Reviewer #1 (Remarks to the Author):

Thanks the authors for addressing my questions in the previous report. The description of the methods has substantially improved. The references are helpful. The results are interesting and novel. I have no additional comments.

Reviewer #2 (Remarks to the Author):

Concerns are properly addressed.

RESPONSE TO REVIEWERS

Reviewer #1 (Remarks to the Author):

Thanks the authors for addressing my questions in the previous report. The description of the methods has substantially improved. The references are helpful. The results are interesting and novel. I have no additional comments.

We thank the reviewer for this feedback.

Reviewer #2 (Remarks to the Author):

Concerns are properly addressed.

We thank the reviewer for this feedback.